# Reticular Chemistry for Optical Sensing of Anions

**DOI:** 10.3390/ijms241713045

**Published:** 2023-08-22

**Authors:** Aasif Helal, Mohd Yusuf Khan, Abuzar Khan, Muhammad Usman, Md. Hasan Zahir

**Affiliations:** 1Interdisciplinary Research Center for Hydrogen and Energy Storage, King Fahd University of Petroleum & Minerals, Dhahran 31261, Saudi Arabia; mykhan@kfupm.edu.sa (M.Y.K.); abuzar@kfupm.edu.sa (A.K.); muhammadu@kfupm.edu.sa (M.U.); 2Interdisciplinary Research Center for Renewable Energy and Power Systems, King Fahd University of Petroleum & Minerals, Dhahran 31261, Saudi Arabia; hzahir@kfupm.edu.sa

**Keywords:** Metal-Organic Frameworks, Zeolitic Imidazolate Frameworks, Covalent-Organic Frameworks, anion recognition, chromogenic, fluorogenic

## Abstract

In the last few decades, reticular chemistry has grown significantly as a field of porous crystalline molecular materials. Scientists have attempted to create the ideal platform for analyzing distinct anions based on optical sensing techniques (chromogenic and fluorogenic) by assembling different metal-containing units with suitable organic linking molecules and different organic molecules to produce crystalline porous materials. This study presents novel platforms for anion recognition based on reticular chemistry with high selectivity, sensitivity, electronic tunability, structural recognition, strong emission, and thermal and chemical stability. The key materials for reticular chemistry, Metal-Organic Frameworks (MOFs), Zeolitic Imidazolate Frameworks (ZIFs), and Covalent-Organic Frameworks (COFs), and the pre- and post-synthetic modification of the linkers and the metal oxide clusters for the selective detection of the anions, have been discussed. The mechanisms involved in sensing are also discussed.

## 1. Introduction

Reticular chemistry is the chemistry of joining pre-selected molecular building blocks with strong interactions to form geometry-guided, precontrived crystalline periodic frameworks that have extensively increased the range of chemical compounds and functional materials. Reticular chemistry has been extensively employed for the last two decades to predict and create a variety of periodically extended structures [1,2,3,4,5,6]. This chemistry enables us to exercise molecular-level control over the matter. Thus, the advantages of reticular chemistry can be highlighted as follows: (a) intricate materials can be constructed with varied components to target explicit structures, (b) resilience in choosing the organic linkers to produce long-lived charge-separated states, (c) effective crystalline topology for selective gas separation on the basis of gating effects, and (d) precise design of the pore interiors by functionalization for selective catalysis. Metal-Organic Frameworks (MOFs), Zeolitic Imidazolate Frameworks (ZIFs), and Covalent-Organic Frameworks (COFs) are the three main examples of reticular chemistry that epitomize the molecular-level manipulation of matter (Figure 1). Consistently, MOFs are fabricated from metal or metal-oxide clusters (Secondary Building Units, SBU) joined by organic linkers through strong metal-carboxyl bonds. This enables extensive variation in the consolidation of the metal ions and the organic linkers to accomplish the pertinent functionalized framework composition and structure [7]. On the other hand, ZIFs are constructed by the reaction of transitional metal ions (Zn^2+^, Co^2+^) with imidazole or imidazole derivatives. The metal centers are coordinated with the nitrogen atom at the 1, 3-position of the imidazolate ligand, forming a metal-imidazole-metal angle of 145°, which is akin to aluminosilicate zeolites but has a bond length larger than zeolites. Thus, ZIFs inherit both the properties of high crystallinity and surface area of the MOF and the high stability of the traditional zeolites [8,9]. Similarly, COFs are constructed from organic building units, comprising entirely light elements of carbon, nitrogen, oxygen, boron, etc., joined by strong covalent bonds and resulting in two- or three-dimensional crystalline porous structures [10]. The disparate competence of reticular chemistry to fabricate tailor-made materials makes this a prime area for application in gas storage and separation [11,12,13,14,15,16,17,18,19,20], water harvesting [21,22,23,24], energy storage [25,26,27,28,29], catalysis [30,31,32,33,34,35], diagnosis and therapy [36,37,38,39,40,41,42,43,44,45,46], and sensing [47,48,49,50,51,52,53,54].

Anions play a pivotal role in diversified biological milieus [55] and environmental systems [56,57,58]. The discharge of toxic anions such as chromate, cyanide, phosphate, halides, permanganate, etc., from industrial and other anthropogenic activities causes severe environmental problems and endangers human health [59,60]. Thus, fabrications of optical (colorimetric/fluorescent)-based chemosensors for anions recognition have acquired a wide interest in the current research due to their high selectivity, sensitivity, and simplicity in detecting meager concentrations of anions. Broadly, most optical sensors consist of a receptor for the selective binding of anions and a signaling unit or reporter to produce discernible optical signals upon binding with specific anions. Based on the design of the sensors, the nature of optical sensing can be categorized into the following groups: (a) the receptor can be integrated with the reporter through the conjugated π bonding or coordination bond in such cases the optical signals are produced by the ligand to metal charge transfer (LMCT) upon anion binding [61]; (b) the receptor for the anion is bonded to the signaling unit through a short spacer and in this the optical signals originate from photo-induced electron transfer (PET), excimer/exciplex formation, photo-induced charge transfer (PCT), fluorescence resonance energy transfer (FRET) and excited-state proton transfer (ESPT) [62,63]; and (c) optical signals can also originate from the irreversible reaction between the anion and the chemosensor commonly known as chemodosimeter type sensing [64], (d) the indicator-displacement approach (IDA) where the anion binding site of the receptor is first occupied by the signaling unit through reversible non-covalent interactions. The addition of anions removes the signaling unit from the receptor, resulting in a change in the optical signals [65].

Some of the advantages of reticular chemistry in the optical sensing of anions as compared to other methods of sensing are:(1)Sensitivity—the presence of sustainable pores enhances guest-host interactions and pre-accumulates the anions, which increases detection sensitivity;(2)Electronic tunability—the electronic properties can be finely tuned by varying the degree of conjugation of the organic linkers, which changes the HOMO-LUMO band gap and thereby tunes the absorption and emission properties;(3)Selectivity—functional groups within the framework, such as Lewis acidic or hydrogen bonding sites in the ligands or varying the nature of the metals in the clusters (in the case of MOFs and ZIFs), can further promote binding of the preferred anion and thereby improve the selectivity of detection;(4)Structural recognition—structure-property correlations and anion-sensor interactions may be thoroughly investigated due to the highly ordered crystalline nature, which makes it easier to characterize and identify structures precisely;(5)Strong emissions—incorporating organic linkers into a rigid framework can minimize the non-radiative relaxation caused by the free rotation and vibration of the linkers, thereby enhancing the emission strength, e.g., aggregation-induced emission (AIE);(6)Thermal and chemical stability—the thermal and chemical stability of these materials is relatively high, allowing them to retain their crystalline structure at elevated temperatures. Thus, they can preserve their optical properties (absorbance and fluorescence) at relatively high temperatures, allowing them to be utilized when binding to a particular anion that necessitates an elevated temperature.

The most important challenge when employing reticular chemistry in the synthesis of anion receptors is to achieve selective binding to a specific anion. When designing small-molecule anion receptors, size and shape matching between the receptor and anion are often implemented to achieve anion selectivity. Based on this concept, it is difficult to develop reticular chemistry-based anion receptors due to the major obstacles in creating a well-organized anion-binding cavity. This problem of poor anion selectivity has been solved to some extent by adjusting the hydrophilic/hydrophobic characteristics of the anion binding sites to correspond with the charge density of the target anion. This review tried to summarize the application of reticular chemistry (MOFs, ZIFs, and COFs) in the optical (chromogenic and fluorogenic) sensing of anions. The anion assay applications of MOFs, ZIFs, and COFs will be investigated individually to provide a useful comparison of the performance and accomplishments of these structures.

## 2. MOFs for Optical Sensing of Anions

A Metal-Organic Framework can act as an effective chromogenic or fluorogenic sensor for anions when it has a binding part for the anions and a signaling part that produces the change in the emission or the absorbance of the sensor. These two parts can be introduced in the MOF during the synthesis of the linker (pre-functionalized MOF), can already be already present in the MOF as its intrinsic property (non-functionalized MOF), or can be introduced after the synthesis of the MOFs (post-functionalized MOFs).

### 2.1. Pre-Functionalized MOFs

Yang et al. reported the synthesis of an intrinsically fluorescent amino derivative of UiO-66 for the detection of the phosphate anion [66]. In this work, the fluorescence of the free linker 2 amino-terephthalic acid (BDC-NH_2_) was quenched on incorporation into the UiO-66 framework due to LMCT. On the introduction of phosphate ions, which have a high affinity towards the Zr-O nodes of the UiO-66-NH_2_, selective binding with the Zr-O clusters weakens the LMCT between the Zr-O nodes and the BDC-NH_2_. As a result, the original fluorescence of BDC-NH_2_ is proportionately recovered depending on the amount of phosphate added. The detection limit was found to be 1.25 μM with high selectivity as compared to halide, sulfate, carbonate, and nitrate anions. In another study, heterometallic MOFs (SmZn(abtc)) and (TbZn(abtc)) [H_4_abtc = 3,3′, 5,5′-azo benzene tetracarboxylic acid] were synthesized. Among these, (TbZn(abtc)) was found to be very effective in the sensing of nitrite by fluorescence quenching [67]. The mechanism of the fluorescence quenching can be explained as nitrite being an efficient excited state quencher of luminescence, causing inner-sphere complexation with the terbium ion that results in an electron exchange energy transfer that results in fluorescence quenching. A chromate ion sensor was reported by Xiao et al. based on Zinc and 4′-[4, 2′; 6′, 4″]-terpyridin-4′-yl-biphenyl-4-carboxylic acid as linkers (Figure 2a,b) [68]. This Zn-MOF showed high sensitivity and selectivity with a low detection limit of 10^−8^ M and very fast quenching and regeneration ability up to six cycles (Figure 2c,d).

The change in the fluorescence was visible to the naked eye and was due to the strong binding of the chromium of the chromate ion with the pyridine nitrogen. From the UV-vis absorption spectra of the chromate ions, it seems that there is an energy overlap between the excitation wavelength of the MOF and the absorption spectra of the chromate ions. Thus, the excitation energy of the framework is absorbed by the chromate ion, resulting in the inhibition of the energy transfer between the ligand and the zinc nodes of the MOF, leading to fluorescence quenching. A highly luminescent cadmium-based mixed linker MOF of molecular formula [Cd_2.5_(PDA)(tz)_3_] {PDA = 1,4−phenylenediacetate and tz = 1,2,4−triazolate} was employed as an efficient sensor for the recognition of the iodide ion in aqueous medium (Figure 3a). The iodide ion caused a highly selective and sensitive quenching of 93% of the Cd-MOF, with a detection limit of 80 ppb in water [69]. The unsaturated cadmium sites are occupied by the iodide ions through the soft-soft acid-base interaction, resulting in the absorption of the excitation energy of the linkers by the iodide ion, which inhibits the charge transfer from the linker to the cadmium clusters, leading to quenching (Figure 3b). Another halide sensor was reported by Zhu et al. [70], in which UiO-66-NH_2_ was employed as a fluoride sensor (Figure 3c). The selective F^−^ anion–enhanced fluorescence was due to the hydrogen bond formation between the NH_2_ of the MOF and fluoride anion, which causes enhanced electron transfer from the linker to the Zirconium oxide clusters. The detection limit for F^−^ anion was found to be 0.229 mgL^−1^ with high selectivity as compared to halide, sulfate, carbonate, phosphate, and nitrate anions (Figure 3d).

Li et al. reported the synthesis of a water-stable polyrotaxane Zn-based mixed linker MOF of 2-amino-5-sulfobenzoic acid (H_2_afsba) and 1,4-bis(triazol-1-ylmethyl) benzene (bbtz) {[Zn(afsba)(bbtz)_1.5_(H_2_O)_2_]·2H_2_O}_n_. This Zn-MOF was found to be potentially highly selective and sensitive toward the detection of chromate ions (CrO_4_^2−^/Cr_2_O_7_^2−^) in the presence of different anions (Figure 4a). The quenching of the fluorescence was due to the strong binding of the chromate with the linkers through the hydrogen bond between the oxygen of the chromate ions and the NH_2_ of the linker and the coordination of the chromium with the triazole group. This results in both fluorescence resonance energy transfer (FRET) and photoinduced electron transfer (PET) from the excited MOF to the Cr(VI) anions, leading to quenching (Figure 4b). The detection limit was found to be 0.22 ppm/0.26 ppm for CrO_4_^2−^/Cr_2_O_7_^2−^ [71]. Similarly, Zhuang and coworkers fabricated a three-dimensional copper-based MOF with the molecular formula of [Cu_2_(tpt)_2_(tda)_2_].H_2_O (tpt = 2,4,6-Tri(pyridin-4-yl)-1,3,5-triazine, H_2_tda = 2,5-thiophene dicarboxylic acid) for highly sensitive and selective detection of chromate ions in acidic media [72]. The binding of the chromium of the CrO_4_^2−^ ion with the nitrogen and sulfur of the linkers results in the overlap of the absorption band of the chromate and the emission band of the framework (FRET). This indicates that the excitation energy of the linker is significantly absorbed by the CrO_4_^2−^ ion, leading to 96% quenching. Moreover, the detection limit was found to be 1.6 × 10^−5^ M indicating high sensitivity and selectivity in the detection of CrO_4_^2−^ ions.

Helal et al. reported the synthesis of UiO-66-NH-BT, a UiO-66 framework containing benzotriazole functionalized dicarboxylate struts, as a very selective and ultrasensitive chromium oxyanions in aqueous media [74]. It showed a detection limit of 280 ppb for Cr_2_O_7_^2−^ and 47.7 ppb for CrO_4_^2−^ anions. The quenching constants (K_sv_) for Cr_2_O_7_^2−^ and CrO_4_^2−^ were found to be 3.9 × 10^3^ and 6.7 × 10^3^, respectively. The covalently bonded benzotriazole moiety with the UiO-66 framework not only produces an emission peak at 491 nm but also acts as an intrinsic binding site for anions. Another thiazolothiazole fluorophore incorporated Zn-MOF [Zn_2_(TzTz)_2_(BDC)_2_]·2DMF, (TzTz = 2,5-di(4-pyridyl)thiazolo [4,5-d]thiazole and BDC = terephthalic acid), was used for the highly selective and reversible detection of Cr_2_O_7_^2−^ and MnO_4_^−^ anions by Safaei and coworkers (Figure 4c) [73]. The sensor showed excellent sensitivity, selectivity, and recyclability in the detection of the Cr_2_O_7_^2−^ and MnO_4_^−^ anions, with the limit of detection being 4 μM for both anions (Figure 4d). From the UV-vis spectra of the only Cr_2_O_7_^2−^ and MnO_4_^−^ anions it was found that there is a maximum overlap with the excitation spectrum of the MOF, and as a consequence, efficient energy transfer allows Cr_2_O_7_^2−^ and MnO_4_^−^ anions to have maximum fluorescence quenching. Gogoi et al. prepared a DUT-52 MOF with a trifluoroacetamido-functionalized linker for the sensing of the cyanide ion with fluorescent enhancement (Figure 5a) [75]. The fluorescence titration experiments exhibited a highly sensitive and selective turn-on behavior of the modified MOF interacting with the cyanide ion (Figure 5b). The detection limit of the probe was found to be 0.23 μM and it showed anti-interference ability in the presence of different anions (Figure 5c). The probe was also effective in real-world water samples from taps, drinking water, lakes, and rivers. The DUT-52 on functionalization with electron-withdrawing trifluoroacetamide weakens the emission intensity due to photoinduced electron transfer (PET). On interaction with the cyanide ion, the trifluoroacetamide group undergoes nucleophilic addition by the cyanide ion to form the cyanohydrin adduct, thus converting into a strong electron-donating group. The PET process is inhibited, resulting in an enhancement in the fluorescence emission. Jindal and coworkers prepared a Zn-MOF with zwitterionic linkers (bisimidazoletetracarboxylic acid) to mimic the naturally occurring amino acids, where the imidazole and the carboxylic acid behave as the zwitterions (Figure 5d) [76]. This MOF showed a 120% enhancement in fluorescent interaction with dihydrogen phosphate (H_2_PO_4_^−^) with a detection limit of 0.13 ppm. This MOF exhibited high sensitivity, selectivity, and anti-interference ability with different anions F^−^, I^−^, Br^−^, AcO^−^, PF_6_^−^, OH^−^, BF_4_^−^, SO_4_^−^, TfO^−^, ClO_4_^−^, NO_3_^−^, HSO_4_^−^, and HCO_3_^−^. The H_2_PO_4_^−^ is an amphiphilic anion that has both hydrogen bond donor and acceptor properties (Figure 5e). This results in a strong hydrogen bonding interaction between the H_2_PO_4_^−^ and the zwitterionic MOF (carboxylate and the protonated imidazolium ion), and as a consequence, 2D metal-organic nanosheets are formed, which results in an immense enhancement of fluorescence (Figure 5f). Zhu et al. reported three MOF-based sensors for the dichromate (Cr_2_O_7_^2−^) ions [77]. {[Zn_3_(BTEC)_2_(H_2_O)(4-BCBPY)].(H_2_O)}n (**1**–**3**) (BTEC^4−^ = 1,2,4,5-benzene tetracarboxylic acid anion, 4-BCBPY^2+^ = 1,1′-bis(4-cyanobenzyl)-4,40-bipyridinium dication) showed fluorescence quenching in the presence of Cr_2_O_7_^2−^. The response was highly selective and sensitive for Cr_2_O_7_^2−^, with a detection limit of 3.28, 7.69, and 10.40 μM, respectively. The mechanism of the color change can be attributed to the structural change and the competitive absorption between the three compounds and Cr_2_O_7_^2−^. In a separate work, a Zn-MOF, {[H_2_N(Me)_2_]_2_[Zn_5_(L)_2_(OH)_2_]·3DMF·4H_2_O}*_n_* was prepared for the selective and sensitive detection of Cr_2_O_7_^2−^ [78]. The detection limit was calculated as 1.86 × 10^−4^ M with a complete quenching of the emission of MOF (Table 1).

### 2.2. Post-Functionalized MOFs

Post-synthetic modifications (PSM) on MOFs, in which chemical transformations or exchanges are performed on pre-synthesized MOF materials, have been discovered to be an effective method for fabricating MOFs based on already existing MOFs. MOFs have been modified with different binding, chromogenic, and/or fluorogenic moieties for the optical sensing of anions. Xu et al. post-synthetically modified the metal nodes of the MIL-124, or Ga_2_(OH)_4_(Hbtc), (H_3_btc = 1,2,4-benzene tricarboxylic acid) by appending Eu^3+^ to the non-coordinated carboxyl group [79]. The resulting MOF was found to be a very selective and sensitive probe for the dichromate ion. The probe exhibited good non-interfered selectivity towards Cr_2_O_7_^2−^ ions with a detection limit of 0.15 μM. The PSM of the metal nodes of the MIL-121(Al(OH)(H_2_btec)H_2_O, H_2_btec = benzene tetracarboxylic acid) was performed by attaching Eu^3+^ with the non-coordinated carboxyl group of the Al nodes (Figure 6a) [80]. The resulting Eu^3+^@MIL-121 exhibited highly selective and sensitive fluorescence quenching with the F^−^ and Cr_2_O_7_^2−^ ions. The detection limit of the Eu^3+^@MIL-121 for F^−^ and Cr_2_O_7_^2−^ ions was calculated to be 0.063 μM and 0.054 μM, respectively (Figure 6b). The inhibition of the ligand-to-metal (Eu^3+^) charge transfer due to the binding of the anions was the primary reason for the quenching in the emission spectra. Dalapati et al. used the UiO-66-NH_2_ framework and post-synthetically modified it with 1-pyrenecarboxaldehyde [81]. This pyrene-tailored MOF displayed a three-fold increase in fluorescence emission due to the formation of the pyrene excimer within the MOF framework. This functionalized MOF exhibited a turn-on fluorescent enhancement and a prominent ratiometric blue shift on binding with the F^−^ and H_2_PO_4_^−^ with a detection limit of 8.2 × 10^−7^ M and 7.3 × 10^−7^ M, respectively.

The extremely selective and sensitive detection of the anions with fluorescence enhancement was mainly due to the hydrogen bond between the imine hydrogen of the pyrene-tailored MOF and the anions that resulted in the static pyrene excimer formation. In another work with UiO-66-NH_2_, imidazole-2-carboxyaldehyde was appended with the MOF post-synthetically (Figure 7a) [82]. This PSM of the UiO-66-NH_2_ helps in the fabrication of a highly efficient sensor for the detection of S_2_O_8_^2−^ with a detection limit of 8.63 μM. The presence of S_2_O_8_^2−^ selectively quenched the fluorescence of the MOF due to the hydrogen bonding of the anion with the acidic hydrogen of the imine bond and the imidazole. The strong binding reduces the energy transfer efficiency between the ligand and the metal (Zr) nodes of the framework, resulting in quenching (Figure 7b). In 2018, post-synthetic modification was reported in MOF, MIL-68 (In)-NH_2_ with glyoxal to make it a highly selective and sensitive probe for the detection of bisulfite ions (Figure 7c) [83]. The fluorescence emission of the probe is turned on when it interacts with the bisulfite ions, which is not observed in the case of other anions. This fluorescence emission turn-on mechanism can be explained by the fact that the bisulfite undergoes a nucleophilic addition reaction with the free aldehyde of the glyoxal appended to the framework. The resulting free –OH group of the adduct forms a hydrogen bond with the nitrogen of the imine, thus inhibiting the rotation of the C=N bond, which leads to the enhancement of fluorescence (Table 1).

### 2.3. Non-Functionalized MOFs

A lanthanide-based 3D Eu-BTB framework was reported by Xu et al. for the highly selective detection of PO_4_^3−^ ion by complete quenching of fluorescence as compared to any other anions such as PO_4_^3−^, F^−^, Cl^−^, Br^−^, I^−^, N_3_^−^, NO_3_^−^, OAc^−^, SCN^−^, IO_3_^−^, BF_4_^−^, ClO_4_^−^, SO_3_^2−^, SO_4_^2−^, CO_3_^2−^, C_2_O_4_^2−^, and P_2_O_7_^4−^ (Figure 8a,b) [84]. The strong affinity and complete quenching were due to the strong binding of the phosphate ion with the Eu-O clusters, which reduces the antenna effect, resulting in emission quenching with a detection limit of 10^−5^ mol/L. Furthermore, another Yttrium-based 1,3,5-benzene tricarboxylic acid MOF doped with Europium was prepared for the sensing of the chromate ions with high selectivity, sensitivity, and detection limit of 0.04 μM and 0.03 μM for Cr_2_O_7_^2−^ and CrO_4_^2−^, respectively [85]. Bhattacharjee et al. synthesized a terbium-based MOF by complete transmetalation of a Ba-MOF [H_2_N(CH_3_)_2_][Ba(H_2_O)(BTB)] to form Tb(H_2_O) (BTB) (BTB = 1,3,5-benzenetribenzoic acid) [61]. This Tb-MOF displayed enhanced fluorescence and acted as a selective and sensitive probe for the phosphate anion. The increase in photoluminescence was due to the antenna effect between the Tb–O clusters and the organic linker. Due to the affinity of the phosphate ions towards the Tb–O clusters, the fluorescence is quenched on binding with the anions, with a detection limit of 35 μM. The quenching in the emission occurred due to the inhibition of the antenna effect on binding with the phosphate ions. A cationic Eu-MOF with the formula [Eu_7_(mtb)_5_(H_2_O)_16_].NO_3_ (H_4_mtb = 4-[tris(4-carboxyphenyl)methyl] benzoic acid) was prepared for the sensitive and selective detection of chromate anion in the presence of other anions in drinking water and the natural water system [86]. The detection limit of this sensor for deionized water, lake water, and seawater is calculated to be 0.56 ppb, 2.88 ppb, and 100 ppb, respectively. The quenching in the fluorescence was due to the inhibition of the antenna effect of the MOF skeleton. This framework was highly selective to the chromate ion even in the presence of other anions such as F^−^, Cl^−^, NO_3_^−^, CO_3_^−2^, SO_4_^−2^, BO_3_^−^, IO_4_^−^, and NO_2_^−^ or in the presence of environmentally abundant cations such as Na^+^, Sr^2+^, Al^3+^, Ca^2+^, Cu^2+^, Mg^2+^, and Zn^2+^. Zirconium-based MOF NU-1000 prepared by Lin and coworkers showed high selectivity for dichromate ions as compared to other anions and had a detection limit of 1.8 μM (Figure 9a) [87]. The fluorescence quenching in the probe on binding with the dichromate was attributed to the electron transfer transitions of the dichromate that reduced the energy transfer between the π and π^*^ orbitals of the ligand of NU-1000. The quenching with the probe slowly takes place with the increase in the chromate concentration (Figure 9b). Another chromate ion sensing Tb-MOF based on *p*-terphenyl-3,3″,5,5″-tetracarboxylic acid as a ligand was prepared by Yu et al. [88]. The lanthanide-based MOF emission was completely quenched on binding with dichromate anion even in the presence of other anions such as Ac^−^, H_2_PO_4_^−^, Cl^−^, CO_3_^2−^, Br^−^, I^−^, SCN^−^, SO_4_^2−^, and NO_3_^−^. Li et al. reported a Cd-MOF [Cd_3_(cpota)_2_(phen)_3_]_n_.5nH_2_O H_3_cpota = 2-(4-carboxy phenoxy)terephthalic acid and phen = 1,10-phenanthroline as a probe for the detection of the chromate ions in water [89]. The detection limit of the MOF for the probing of Cr_2_O_7_^2−^ and CrO_4_^2−^ was reported to be 0.37 μM and 0.418 μM, respectively, in the aqueous medium. The mechanism of fluorescence quenching on the binding of the probe with the anion is due to the decrease in the energy transfer between the p-p^*^ orbitals of the linker caused by the electron transfer from the donor (the organic linker) to the acceptor (Cr_2_O_7_^2−^ and CrO_4_^2−^). Chen et al. reported a Eu^3+^ TCPB (1,2,4,5-tetrakis(4-carboxyphenyl)-benzene)-based MOF for the ratiometric fluorescence sensing of phosphate ions in aqueous medium (Figure 9c) [90]. The lanthanide-based probe was found to produce an increased ratiometric change in fluorescence only in interaction with the phosphate ion. The addition of any other anion neither produces such a change with the probe nor interferes with the sensing of the phosphate by the probe. The detection limit was found to be 0.145 mM. On exploring the sensing mechanism, it was concluded that as the phosphate ion binds with the Eu^+3^ cluster, the characteristic lanthanide emission peaks decrease while the emission peak of the ligand is enhanced due to the transmission of energy from the Eu^+3^ to the ligand (Figure 9d). (Table 1). Thus, from the above discussion, it is evident that the proper post- and pre-synthetic modifications of the linkers and the metal clusters help in the sensitive and selective detection of anions.

## 3. ZIFs for Optical Sensing of Anions

ZIFs are usually functionalized with the binding group or can be post-synthetically modified to act as an effective anion sensor. In some cases, they are incorporated with signaling units or act as a support for the development of anion sensors. Liu et al. changed the luminescent property of ZIF-90 and used it as a sensor for the detection of chromate ions [91]. The sensor showed strong selectivity, sensitivity, and anti-interference ability in the sensing of the chromate ion by quenching the fluorescence of the ZIF-90 in the presence of other anions NO_3_^−^, C_2_O_4_^2−^, CO_3_^2−^, Br^−^, F^−^, Cl^−^, I^−^, PO_4_^3−^, CrO_4_^2−^, HCO_3_^−^, NO_2_^−^, and S^2−^. From the UV-vis spectrum, it is clear that there is an overlap in the absorption of the analytes and the probe. Thus, the chromate competes with the ligand in the absorption of the excitation energy, thus decreasing the energy transfer from the ligand to the metal SBU of the ZIF-90, resulting in the quenching of the fluorescence. Another work on post-synthetic modification of highly luminescent ZIF-90 with a dicyanovinyl group was undertaken by Karmakar and coworkers (Figure 10a) [92]. The dicyanovinyl group is known to be a proficient cyanide ion receptor. Thus, its introduction in the ZIF-90 makes it a highly sensitive and selective probe with anti-interference ability for the detection of cyanide ions in the presence of different anions such as F^−^, Cl^−^, Br^−^, SCN^−^, NO_3_^−^, NO_2_^−^, and N_3_^−^ (Figure 10b). The fluorescence of the probe is quenched with the addition of the cyanide ion, and the limit of detection was calculated to be 2 μM. The mechanism behind the quenching is the loss in conjugation due to the nucleophilic addition of the cyanide group, which causes 90% quenching of the fluorescence of the probe. Li and coworkers reported a Eu^3+^ complex functionalized Fe_3_O_4_ nanoparticle (Eu-BBA-PEG-DBA-Fe_3_O_4_) encapsulated in the ZIF-8 for the simultaneous detection of the ClO^−^ and SCN^−^ [93]. The Eu^3+^ complex of BBA-PEG-DBA (BBA = 2-benzoylbenzoic acid, DBA = 3,4-dihydroxybenzylamine, and PEG = Polyethylene glycol) was first prepared, followed by stirring with Fe_3_O_4_ nanoparticles to form the nanocomposite Eu-BBA-PEG-DBA-Fe_3_O_4_. This nanocomposite is incorporated by the in situ method during the synthesis of ZIF-8 (Figure 10c). The core-shell-like nanocomposite PDA-Eu-BBA-PEG-DBA-Fe_3_O_4_@ZIF-8 exhibited the characteristic emission of the lanthanides, which was selectively quenched by the ClO^−^ in an aqueous system with a detection limit of 0.133 nM, and the red color of the solution was turned colorless (Figure 10d). This quenched PDA-Eu-BBA-PEG-DBA-Fe_3_O_4_@ZIF-8- ClO^−^ composite was used for the selectively sensitive detection of SCN^−^ in water. Upon addition of the SCN^−^ the quenched fluorescence was enhanced and regained, with the color of the solution changing from colorless to red. The detection limit for the sensing of the SCN^−^ was calculated to be 0.204 nM. The mechanism by which the fluorescence turns off and on is due to the strong affinity of the ClO^−^ towards Eu^3+^ that releases the nonfluorescent PDA, thus quenching the overall system. The addition of the SCN^−^ ion triggers an oxidation-reduction reaction between the two analytes (ClO^−^ and SCN^−^) that makes Eu^3+^ coordinate with the PDA, again restoring the fluorescence. This probe was effective in the detection of ClO^−^ and SCN^−^ in tap water as well as river water and can be reused for five cycles. Chen et al. reported the synthesis of MAPbBr_3_@ZIF-8 composite (MAPbBr_3_ = Methylammonium bromide and lead bromide) for the sensing of hypochlorite (ClO^−^) in the aqueous medium [94]. The MAPbBr_3_ nanoparticles were incorporated in the ZIF-8 by the simple in situ stirring of a mixture of lead bromide (PbBr_2_), methylammonium bromide (MABr), zinc acetate, and 2-methylimidazole. The fluorescence of the MAPbBr_3_@ZIF-8 composite was completely quenched with the addition of the ClO^−^ and it showed excellent stability and sensitivity in the detection of the ClO^−^ with a detection limit of 31.9 nM. The composite also showed high selectivity and anti-interference ability with different anions such as HCO_3_^−^, SO_4_^2−^, Cl^−^, SiO_3_^2−^, CO_3_^2−^, NO_3_^−^, and HSO_3_^−^. The probe exhibited excellent sensing properties for ClO^−^ in real environments with deionized water, tap water, and lake water. The selective quenching mechanism of the probe was explained with the help of PXRD, fluorescence lifetime measurement, and FT-IR. The ClO^−^ ion penetrates the channels within the ZIF-8 and forms hydrogen bonds with the N-H of the MAPbBr_3_ nanoparticles that result in the electron transfer between the ClO^−^ and the nanoparticles; as a consequence, the fluorescence is quenched (Table 1). ZIFs act as effective anion sensors due to the presence of narrow channels and post-synthetic functionalization. Sometimes ZIFs are also used as a support for certain fluorogenic and chromogenic moieties that can selectively detect anions.

## 4. COFs for Optical Sensing of Anions

The robust structure and large conjugated fluorescent chromophores, such as phenyl, naphthalenyl, pyrenyl, perylenyl, triazine, and triphenyl-benzene, are important properties of COFs. Additionally, conjugated linkages such as imines and olefins can expand the conjugated structure of COFs. Thus, COFs can act as chromogenic and fluorogenic anion sensors. Li et al. synthesized a luminescent highly conjugated porous covalent organic framework using a carbon–carbon coupling reaction of triarylboron [95]. The COF was synthesized by using Suzuki–Miyaura cross-coupling of tris(4-bromo-2,6-dimethylphenyl)borane and tris(4-dihydroxyboranylphenyl)amine (TBPA) in the presence of Pd(0) as the catalysts (Figure 11a). This COF BCMP-3 displayed a high emission band at 488 nm and a quantum yield of 18% in the solid state. The emission is attributed to the charge transfer between the triarylamine donor and triaryl borane acceptor. When investigated with different anions such as F^−^, Cl^−^, Br^−^, NO_3_^−^, HSO_4_^−^, and PF_4_^−^ the COF exhibited a change in the color observed by the naked eye in the THF solution due to the decrease in the charge transfer band (Figure 11b). The same change was observed in fluorescence, where the green fluorescence of the COF changed to blue with the addition of only fluoride ions. This ratiometric change in the emission was due to the binding of the fluoride with the boron sites that inhibit the charge transfer from the triarylamine donor and triaryl borane acceptor. Ordered π structures in the COF are important in the development of fluorescent-based sensors. Moreover, in hydrazone-based COFs, deprotonation of the N-H will inhibit the quenching profile of the COF. Based on this perception, a hydrazone-linked TFPPy-DETHz-COF by condensation of 1,3,6,8-tetrakis(4-formylphenyl)pyrene (TFPPy) and 2,5-diethoxyterephthalohydrazide (DETHz) [96]. The COF displayed a green-yellow fluorescence at 540 nm with an absolute fluorescence quantum yield of 4.5%. The addition of different anions to the COF solution in THF, but only fluoride ions, causes a distinct increase in the fluorescence of 3.8-fold and an absolute fluorescence quantum yield of 17%. The highly selective and sensitive detection of fluoride ions with the COF has a detection limit of 50.5 ppb. The enhancement in the fluorescence is attributed to the acid-base reaction between the N-H of the hydrazone and the fluoride ion, leading to deprotonation that inhibits the electron transfer from the hydrazone linkage to the pyrene moiety.

Another fluoride ion sensor was prepared by Su et al. based on peroxidase mimics [97]. In this work, a 2D covalent triazine framework was prepared with the incorporation of Fe to form Fe-CTF. This COF was then applied for the sensitive and selective colorimetric detection of fluoride ions. The Fe-CTF showed anti-interference ability with different anions in the detection of fluoride ions and has a detection limit of 5 nM. The colorimetric change was mainly due to the stable complexation of the fluoride ion with the Fe to form FeBr_3_, which detaches the Fe from the framework of the CTF and in turn inhibits the charge transfer during the peroxidase-like catalytic activity (Figure 11c). Li and coworkers reported a 3D COF of Tetra(p-aminophenyl)methane Bis(tetraoxacalix[2]arene[2]triazine) for the fluorescence sensing of CrO_4_^2−^, Cr_2_O_7_^2−^, and MnO_4_^−^ ions with excellent sensitivity, selectivity, and recyclability [98]. The fluorescence of the COF was completely quenched in the presence of only CrO_4_^2−^, Cr_2_O_7_^2−^, and MnO_4_^−^ ions. None of the other anions was able to change the fluorescence of the COF, and they exhibited a very low detection limit of 3.43 × 10^−4^, 3.43 × 10^−4^, and 3.20 × 10^−4^, respectively. In order to investigate the mechanism of quenching, the absorption bands of only CrO_4_^2−^, Cr_2_O_7_^2−^, and MnO_4_^−^ ions were compared with the absorption bands of the COF. It was observed that the absorption band of the analytes overlaps the absorption band of the COF. Thus, the analytes inhibit the excitation energy transfer in the COF by absorbing most of the excitation energy, leading to complete quenching of the emission.

Another fluoride ion sensor was reported by Singh et al. based on the exfoliation of COF into Covalent Organic Nanosheets (CONs) that expose more active sites and help in the reduction of fluorescence turn-off phenomena [99]. Based on this, self-exfoliable ionic CONs (DATG_Cl-_iCONs) were prepared by the condensation of triaminoguanidinium chloride (TG_Cl_) with a fluorophore, 2, 5-dimethoxyterephthalaldehyde (DA) (Figure 11d). The high fluorescence properties and the well-exposed active sites made it a very selective sensor for fluoride ions, which causes quenching in the emission on binding with a detection limit of 5 ppb. The selectivity, quenching, and anti-interference ability of the sensor with fluoride ions can be explained by the fact that only strong basic fluoride ions can abstract the acidic proton from the guanidinium ion, which results in neutral guanidine that can transfer electrons to the fluorophore, leading to the quenching in the fluorescence. The quenched fluorescence of the probe is restored by treating it with acid, and thus it can be reused for five cycles. Huang et al. post-synthetically modified the COF TzDa (4,4′,4″-(1,3,5-triazine-2,4,6-triyl)trianiline (Tz) and 1,4-dihydroxyterephthalaldehyde (Da)) with acyl chloride of 2-phenylpropionic acid to produce a highly selective and sensitive sensor for MnO_4_^−^ (Figure 12a) [100]. The fluorescence of the functionalized COF was completely quenched by the addition of MnO_4_^−^ ions; none of the other ions such as F^−^, Cl^−^, Br^−^, I^−^, BrO_3_ ^−^, IO_3_
^−^, SO_4_^2−^, PO_4_^3−^, SCN^−^, CH_3_COO^−^, C_2_O_4_^2−^, CrO_4_^2−^, and Cr_2_O_7_^2−^ were able to cause a significant change in the fluorescence of the COF (Figure 12b). The limit of detection for MnO_4_^−^ was found to be 0.01 mM. The quenching is directly related to the spectral overlap of the analytes and the excitation energy of the COF. A MOF and COF composite was prepared to overcome the aggregation-caused quenching in the COF and detect anions [101]. In this work, UiO-66-NH_2_ was integrated with COF1, formed by condensation of triformylphloroglucinol with phenylenediamine, through the aldehyde of the COF and the amine of the MOF. Due to the strong affinity of the phosphate for the Zirconium cluster, the composite exhibited a new peak in the emission spectrum at 470 nm with the addition of the phosphate ions. The enhancement of the emission peak was unique for only the PO_4_^3−^ ion, and no other anions produce such emission or interfere with the emission of the PO_4_^3−^. The detection limit was calculated to be 0.067 μM, and the probe was practically tested with real water samples. In another work on phosphate sensing by COF, Afshari et al. prepared an imine-linked COF using 1,5-diaminonaphthalene and 2,4,6-tris(4-formylphenoxy)-1,3,5-triazine (Figure 12c) [102]. The fluorescence emission of this COF was significantly quenched by the PO_4_^3−^, CO_3_^2−^, and AsO_4_^3−^ ions, with the maximum occurring in the case of PO_4_^3−^ (Figure 12d). The limit of detection for the PO_4_^3−^ and the CO_3_^2−^ was calculated to be 0.61 × 10^−6^ M and 1.2 × 10^−6^ M, respectively, with recyclability for about five cycles. From the DFT calculations, it is concluded that the quenching of the emission is mainly due to the reduction in the number of excited molecules in the COF- PO_4_^3−^ complex. Wan et al. prepared two different types of COFs (TFHPB-TAPB-COF and TFHPB-TTA-COF) by the Schiff base condensation reaction between the 1,3,5-tris(4-formyl-3-hydroxyphenyl)-benzene (TFHPB) and the 1,3,5-tris(4-aminophenyl)benzene (TAPB) or 4,4′,4″-(1,3,5-triazine-2,4,6-triyl)trianiline (TTA) [103]. The COFs displayed good fluorescence emission due to the formation of intramolecular hydrogen bonds between the imine bonds and the hydroxyl group, resulting in excited-state intramolecular proton transfer (ESIPT). Due to the strong basic nature of the fluoride ion, this ESIPT is inhibited by the deprotonation of the hydroxyl group, resulting in the quenching of the emission in both COFs. The quenching profile of the two COFs was only displayed with the fluoride ion. Anions such as Br^−^, I^−^, HCO_3_^−^, NO_3_^−^, and Cl^−^ did not produce any quenching and did not interfere with the quenching of the fluoride ions. Another fluoride ion sensor was prepared by Wang et al. based on the azine-based COF, called the ACOF [104]. This COF had excellent stability and porosity, with high sensitivity and selectivity towards the fluoride anion. The presence of the hydroxyl group and imine bond results in an ESIPT that induces emission. On binding with the fluoride ion, the ESIPT is inhibited, which results in the fluorescence turning off. The detection limit was calculated to be 2.5 μM (Table 1). The main source of photoluminescence in the case of COF is due to the presence of extended conjugations. The main source of binding with the anions is the hydrogen bonding between the hydrogen of the COF and the anions.

## 5. Conclusions

This review illustrates the contribution of reticular chemistry in the field of selective anion optical sensing, which was accomplished by carefully designing MOFs, ZIFs, and COFs. Reticular chemistry plays an important role in the detection of specific anionic species from a biological, medical, environmental, and industrial standpoint. Reticular chemistry-based materials have certain advantages over other materials, such as sensitivity, selectivity, electronic tunability, structural recognition, strong emission, and thermal and chemical stability. It illustrates how the different reticular chemistry-based materials, MOFs, ZIFs, and COFs, with completely different natures of binding and signaling units, contribute to the optical recognition of anions. The essential factors for selective guest anion recognition are judicious selection of ligand structures and metal ions (in the case of MOFs and ZIFs), incorporation of suitable functional groups in the framework, and pre-and post-synthetic modification of the reticular chemistry materials. The review elaborates on the effect of active metal sites, transmetalation, doping with optically active metals, diverse functionalized ligands with basic and acidic groups, and lanthanides on the sensing efficacy of the MOF. The post-synthetic modifications and the incorporation of the lanthanides in the ZIF structure for the highly sensitive and selective optical sensing of anions with good recyclability have been discussed. The different mechanisms for the optical sensing of anions have been explained, and the cause of high selectivity for certain anions has been highlighted. The nature of anion sensing in the COFs due to the easily tunable structures, porous nature, and diverse aromatic rings endows the COFs with fluorogenic and chromogenic properties. In recent years, considerable effort has been invested in the formation of new MOFs, ZIFs, and COFs with diverse functions. Although reticular chemistry-based materials as sensors have been used to detect a variety of analytes, the emergence of new materials is expected to further expand the analyte detection range. The selectivity and sensitivity will be improved, along with the fast and real-time detection of the analytes. Optical sensors that are portable, wearable, or even implantable are gaining increasing interest. MOFs, ZIFs, and COFs, due to their tunable nature, can act as an anion concentrator as well as a signal generator at the same time. Thus, it can reduce the size and complexity of a sensor and play an important role in the formation of a miniaturized sensing device that can make these materials vital for biomedical applications. This would lead to the development and fabrication of reticular chemistry-based materials as next-generation materials for manufacturing smart devices for anion sensing.

## Figures and Tables

**Figure 1 ijms-24-13045-f001:**
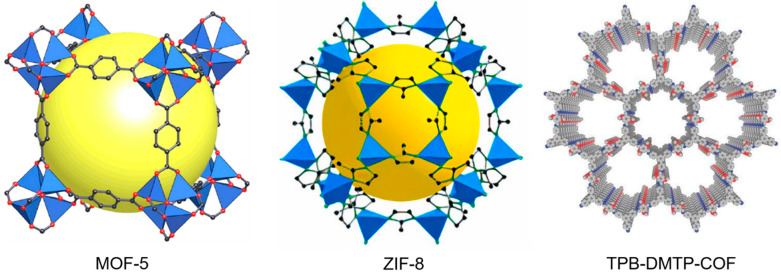
Examples of reticular chemistry.

**Figure 2 ijms-24-13045-f002:**
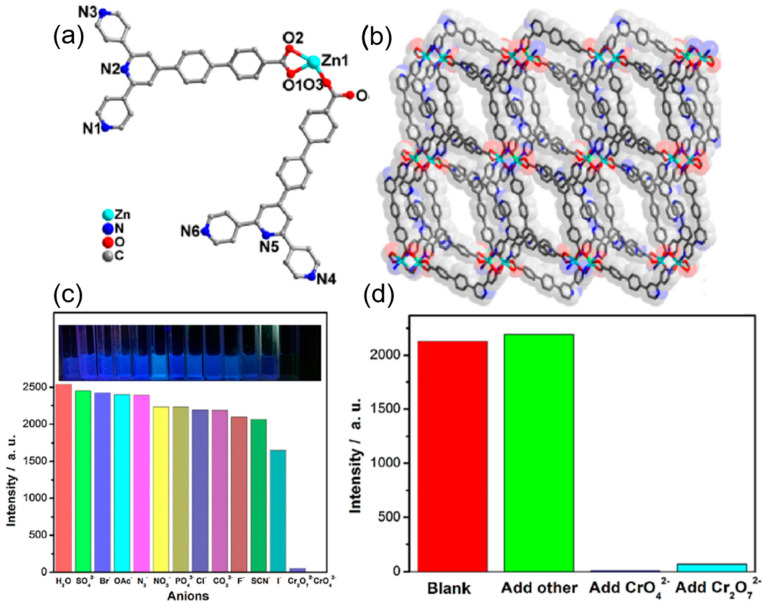
(**a**) The asymmetrical unit of Zn-MOF with hydrogen atoms omitted for clarity. (**b**) The three-dimensional framework of the Zn-MOF. (**c**) The relative intensities at 414 nm for Zn-MOF dispersed in different anions aqueous solutions upon excitation at 358 nm. Inset: the corresponding photographs under the irradiation of 365 nm UV light. (**d**) The relative intensities at 414 nm for Zn-MOF [68]. Copyrights © 2018 Elsevier B.V. All rights reserved. Reproduced with permission from Xiao, J., Sens. Actuators B, published by 2018 Elsevier B.V., 2018.

**Figure 3 ijms-24-13045-f003:**
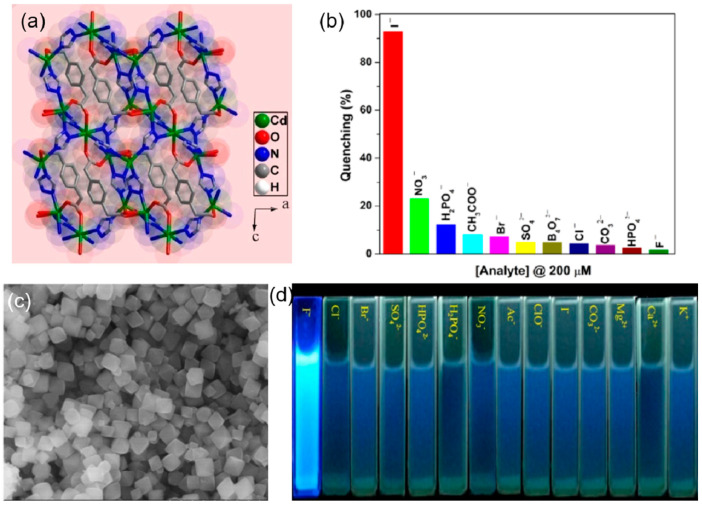
(**a**) The three-dimensional structure of **1 is** formed through the interconnected cages. (**b**) Percentage of luminescence quenching with respect to emission at 290 nm of **1** with 200 μM of different anions [69]. Reproduced with permission from Singh, D.K., J. Photochem. Photobiol. A, published by 2018 Elsevier B.V., 2018. (**c**) SEM images of NH_2_-UiO-66 with a Zr/ATA ratio of 1.04:1. (**d**) Under the excitation of 365 nm, fluorescent images of NH_2_-UiO-66 were immersed into a 3.00 mL aqueous solution containing various ions [70]. Reproduced with permission from Zhu, H., J. Lumin., published by 2019 Elsevier B.V., 2019.

**Figure 4 ijms-24-13045-f004:**
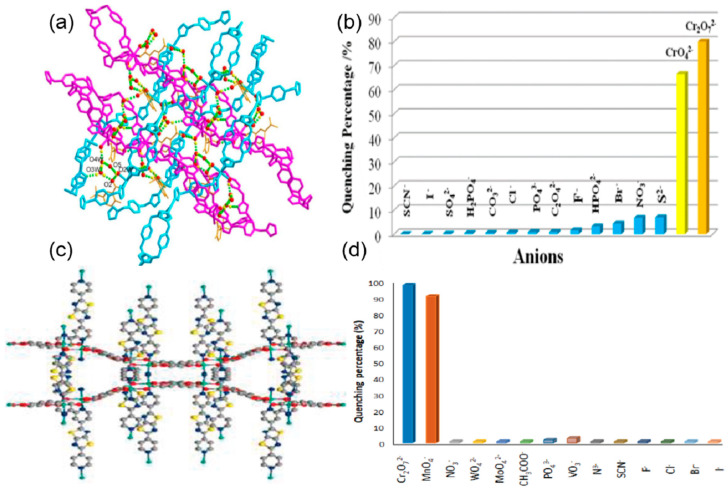
(**a**) A three-dimensional supramolecular network of Zn-MOF is formed through the interlayer hydrogen bonds, involving a hydrogen-bonded cyclic ring. (**b**) The fluorescence quenching percentage is related to the different anions [71]. Reproduced with permission from Li, P.-C., J. Lumin., published by 2019 Elsevier B.V., 2019. (**c**) View of a single network of the Zn-MOF (cyan, Zn; blue, N; red, O; yellow, S; gray, C) along the c axis. (**d**) Selectivity of MOF towards sensing anions at equal concentration by the sensor [73]. Reproduced with permission from Safaei, S., J. Solid State Chem., published by 2021 Elsevier B.V., 2021.

**Figure 5 ijms-24-13045-f005:**
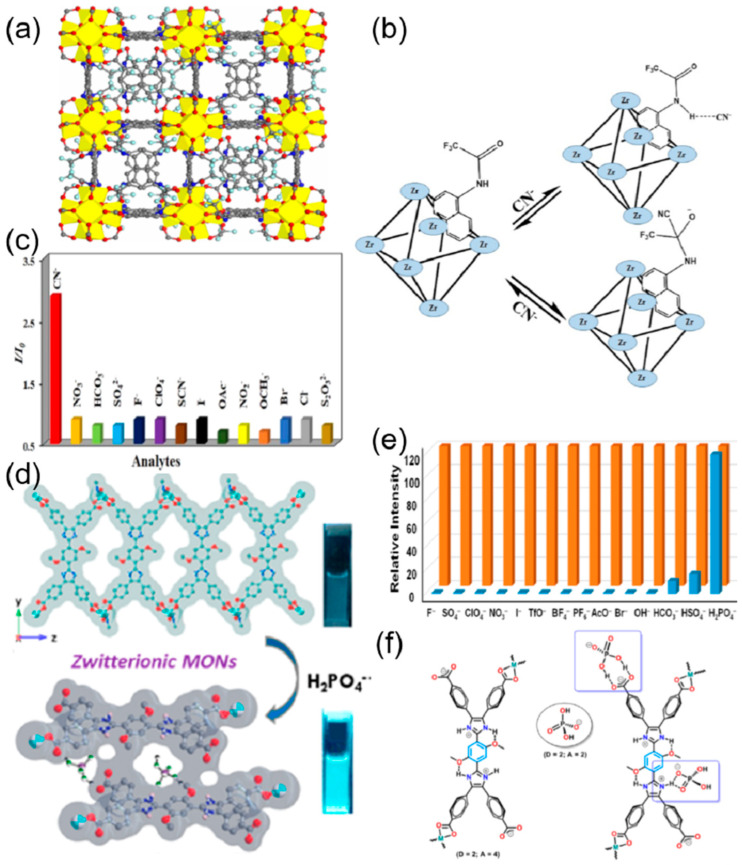
(**a**) Simulated cubic framework structure of DUT-52-NHCOCF_3_ MOF (1) in a ball and stick model. The C, O, N, and F atoms are shown as gray, red, blue, and aquamarine-colored balls, respectively. The Zr clusters are displayed as yellow polyhedra. (**b**) Possible mechanisms of CN− sensing. (**c**) Fluorescence turn-on response of the sensor with different anions in water [75]. Reproduced with permission from Gogoi, C., Inorg. Chem., published by 2021 American Chemical Society, 2021. (**d**) Luminescent 2D Metal−Organic Framework Nanosheets (MONs). (**e**) Fluorescence intensities of 2D MON in the presence of various anionic analytes (orange). 2D-MONs containing H_2_PO_4_^−^ in the presence of other anions (blue). (**f**) Schematic drawing of the structure of the zwitterionic linker and plausible binding of H_2_PO_4_^−^ [76]. Reproduced with permission from Jindal, S., Inorg. Chem., published by 2022 American Chemical Society, 2022.

**Figure 6 ijms-24-13045-f006:**
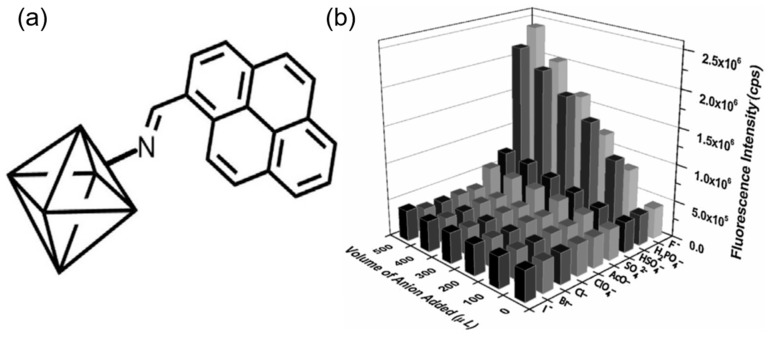
(**a**) UiO-66-NH_2_ functionalized with pyrene. (**b**) Change in the fluorescence intensity of the sensor upon incremental addition of various anions [81]. Reproduced with permission from Dalapati, R., Sens. Actuators B, published by 2017 Elsevier B.V., 2017.

**Figure 7 ijms-24-13045-f007:**
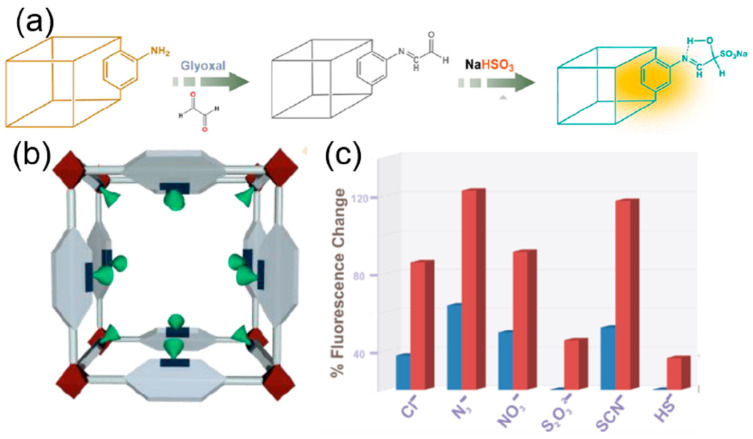
(**a**) Probable mechanism of turn-on response in the presence of HSO_3_^−^ ion. (**b**) PSM of NH_2_-MIL-68(In). (**c**) Increase in fluorescence intensity upon addition of bisulfite ions and other anions to NH_2_-MIL-68(In)@CHO in water [83]. Reproduced with permission from Sen, A., Polyhedron, published by 2018 Elsevier B.V., 2018.

**Figure 8 ijms-24-13045-f008:**
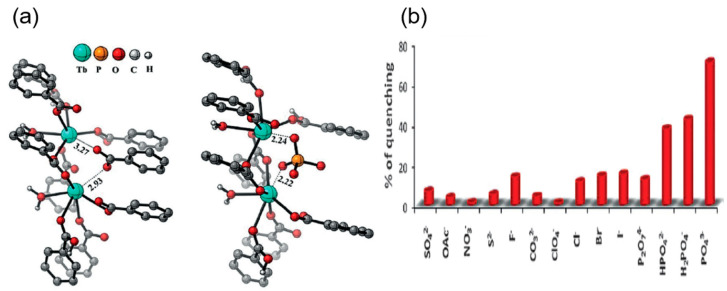
(**a**) The comparison of the strength of binding between the carboxylate and phosphate ions to the metal center in MOF. (**b**) The selective detection of the phosphate anion by the sensor over other anions [86]. Reproduced with permission from Liu, W., ACS Appl. Mater. Interfaces, published by 2017 American Chemical Society, 2017.

**Figure 9 ijms-24-13045-f009:**
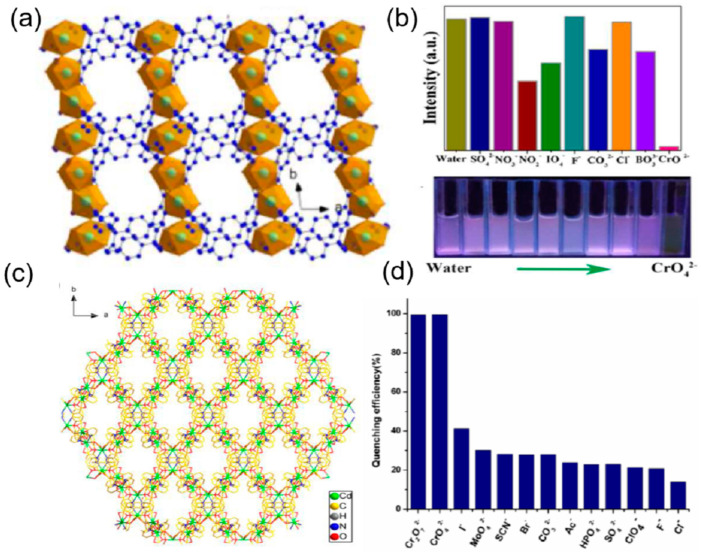
(**a**) A 3D network structure of Eu-MOF viewed along the *c* axis. (**b**) Luminescence intensity of Eu-MOF dispersed into different aqueous solutions of various anions and cations. Luminescence photograph of 1 immersed in different anion solutions [87]. Reproduced with permission from Lin, Z.-J., Inorg. Chem., published by 2017 American Chemical Society, 2017. (**c**) A 3D framework of Cd-MOF. (**d**) Luminescence quenching percentage of Cd-MOF in the presence of different anions [90]. Reproduced with permission from Chen, B.C., Inorg. Chem., published by 2023 American Chemical Society, 2023.

**Figure 10 ijms-24-13045-f010:**
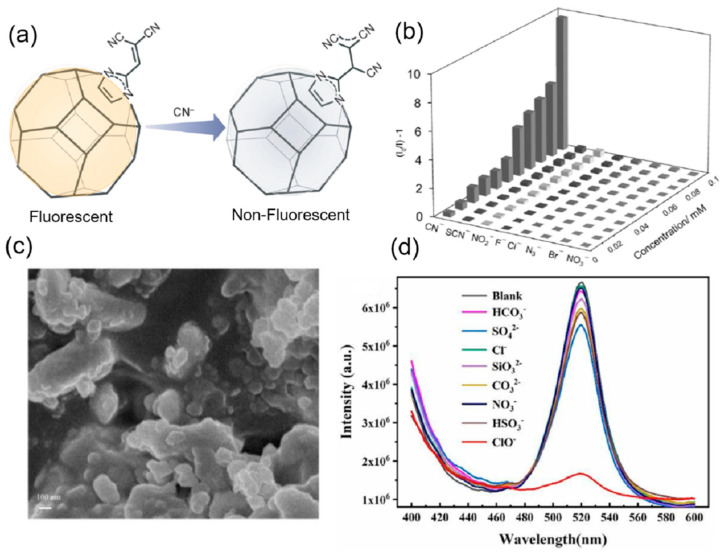
(**a**) Fluorescence modulation of M-ZIF-90 upon addition of CN^−^ ions. (**b**) Fluorescence change of M-ZIF-90 upon addition of other anions (black) followed by the addition of cyanide ions (gray) [92]. Reproduced with permission from Karmakar, A., Chem.—A Eur. J., published by 2015 Wiley-VCH Verlag GmbH & Co. KGaA, Weinheim, 2016. (**c**) SEM images of MAPbBr3@ZIF-8. (**d**) The fluorescence spectra of MAPbBr3@ZIF-8 composites were added with different anions. Inset: under a 365 nm UV lamp, the photographs showed that MAPbBr3@ZIF-8 after adding ClO^−^ solution [94]. Reproduced with permission from Chen, R., Microchem. J., published by 2022 Elsevier B.V., 2022.

**Figure 11 ijms-24-13045-f011:**
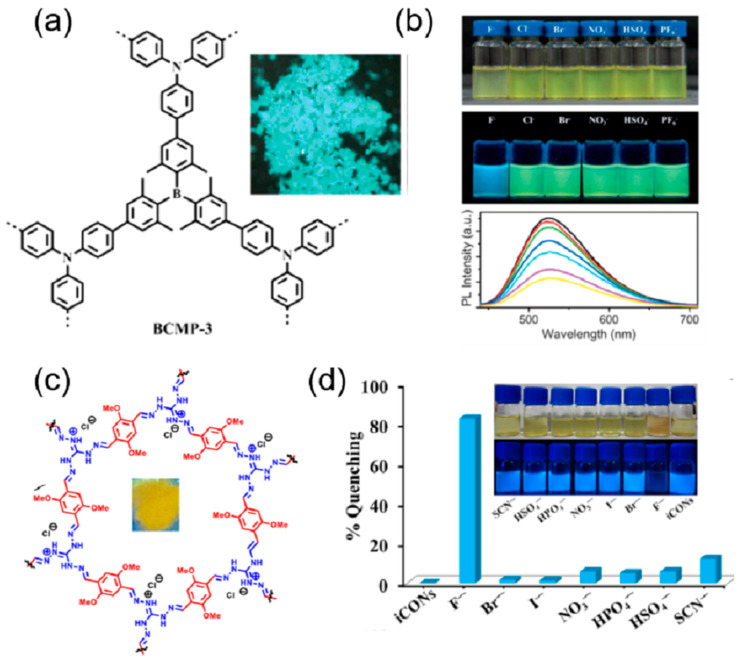
(**a**) Boronic acid-based BCMP-3. (**b**) Photographs showing THF suspensions of BCMP-3 with various anions, the fluorescence of BCMP-3 with anions in THF under UV irradiation at 365 nm, and photoluminescent spectra of BCMP-3 in THF suspensions containing different concentrations of F^−^ [95]. Reproduced with permission from Li, Z., Chem.—A Eur. J., published by 2015 Wiley-VCH Verlag GmbH & Co. KGaA, Weinheim, 2015. (**c**) Ionic COF nanosheets (iCON). (**d**) Percentage fluorescence quenching of iCONs on the addition of various anions [99]. Reproduced with permission from Singh, H., ACS Appl. Mater. Interfaces, published by 2020 American Chemical Society, 2020.

**Figure 12 ijms-24-13045-f012:**
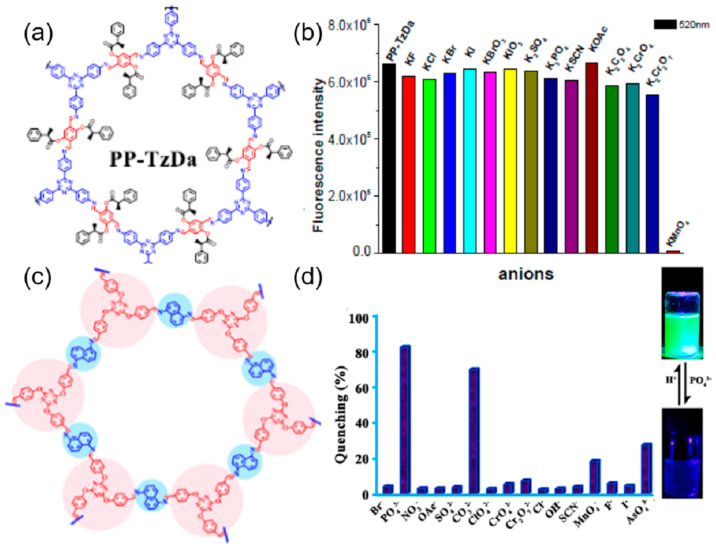
(**a**) Structure of PP-TzDa. (**b**) Response of fluorescence intensity of PP-TzDa at 520 nm after the addition of MnO_4_^−^ or other different anions [100]. Reproduced with permission from Huang, M., Inorg. Chem. Commun., published by 2020 Elsevier B.V., 2020. (**c**) Structure of the IC-COF. (**d**) Percentage fluorescence quenching of IC-COF upon the addition of multiple anions. Change in the fluorescence upon the addition of phosphate ions under UV light [102]. Reproduced with permission from Afshari, M., ACS Appl. Mater. Interfaces, published by 2022 American Chemical Society, 2022.

**Table 1 ijms-24-13045-t001:** Anion sensing properties of MOFs, ZIFs and COFs.

S. N.	Material	Analyte	Binding Constant (M^−1^)	LOD	Surface Area (m^2^g^−1^)	Recyclability (Cycles)	Media	Ref.
Metal-Organic Frameworks (MOFs)								
1	UiO-66-NH_2_	PO_4_^3−^		1.25 μM	671		water	[66]
2	TbZn(abtc)	NO_2_^−^					water	[67]
3	[Zn(tpbpc)_2_]	CrO_4_^2−^ and Cr_2_O_7_^2−^	1.65 × 10^5^ and 1.13 × 10^4^	4.66 × 10^−8^ M and 6.76 × 10^−7^ M		6	water	[68]
4	Cd-MOF	I^−^	1.8 × 10^4^	0.63 µM			water	[69]
5	NH_2_-UiO-66	F^−^		0.229 mg L^−1^	1200		water	[70]
6	[Zn(afsba)(bbtz)_1.5_(H_2_O)_2_]·2H_2_O	CrO_4_^2−^ and Cr_2_O_7_^2−^	2.06 × 10^4^ and 4.42 × 10^4^	0.22 ppm and 0.26 ppm			water	[71]
7	[Cu_2_(tpt)_2_(tda)_2_].H_2_O	CrO_4_^2−^	2.1 × 10^4^	1.64 × 10^−5^ M			water	[72]
8	UiO-66-NH-BT	CrO_4_^2−^ and Cr_2_O_7_^2−^	6.7 × 10^3^ and 3.9 × 10^3^	47.7 ppb and 280 ppb	384	5	water	[74]
9	[Zn_2_(TzTz)_2_(BDC)_2_]·2DMF	Cr_2_O_7_^2−^ and MnO_4_^−^	9 × 10^7^ and 4.8 × 10^3^	4.0 μM		4	water	[73]
10	DUT-52	CN^−^		0.23 μM	1105	2	water	[75]
11	Zn-DMBI	H_2_PO_4_^−^	5.1 × 10^4^	0.13 ppm	72.9		water	[76]
12	Compound 1	Cr_2_O_7_^2−^	9.12 × 10^3^	3.28 μM			water	[77]
13	Compound 2	Cr_2_O_7_^2−^	1.56 × 10^4^	7.69 μM			water	[77]
14	Compound 3	Cr_2_O_7_^2−^	8.60 × 10^3^	10.40 μM			water	[77]
15	{[H_2_N(Me)_2_]_2_[Zn_5_(L)_2_(OH)_2_]·3DMF·4H_2_O}*_n_*	Cr_2_O_7_^2−^	1.455 × 10^4^	1.86 × 10^−4^ μM		5	water	[78]
11	Eu^3+^@MIL-124	Cr_2_O_7_^2−^	6.034 × 10^4^	0.15 µM			water	[79]
12	Eu^3+^@MIL-121	F^−^ and Cr_2_O_7_^2−^	2.07 × 10^3^ and 4.34 × 10^3^	0.063 μM and 0.054 μM	165		water	[80]
13	Pyrene taggedUiO-66-NH_2_	F^−^ and H_2_PO_4_^−^		8.2 × 10^−7^ M and 7.3 × 10^−7^ M			water	[81]
14	UiO-66-NH_2_-IM	S_2_O_8_^2−^	2.883 × 10^3^	8.63 µM	352		water	[82]
15	NH_2_-MIL-68(In)@CHO	HSO_3_^−^		0.047 ppm			water	[83]
16	Eu-BTB	PO_4_^3−^	7.97 × 10^3^	10^−5^ mol/L			water	[84]
17	Y(BTC)(H_2_O)_6_]_n_:0.1Eu	CrO_4_^2−^ and Cr_2_O_7_^2−^	1.18 × 10^3^ and 4.52 × 10^3^	0.03 μM and 0.04 μM	166.04		water	[85]
18	Tb(H_2_O)(BTB)	PO_4_^3−^		35 μM			water	[61]
19	[Eu_7_(mtb)_5_(H_2_O)_16_].NO_3_	Cr_2_O_7_^2−^	3.3 × 10^4^	0.56 ppb	634.5		water	[86]
20	NU-1000B	Cr_2_O_7_^2−^	1.3 × 10^4^	1.8 μM	2288	3	water	[87]
21	[Tb_2_(ptptc)1.5(H_2_O)_2_]_n_	Cr_2_O_7_^2−^					water	[88]
22	[Cd_3_(cpota)_2_(phen)_3_]_n_·5nH_2_O	CrO_4_^2−^ and Cr_2_O_7_^2−^	6.9 × 10^3^ and 1.21 × 10^4^	4.18 × 10^−7^ M and 3.70 × 10^−7^ M			water	[89]
23	{[(CH_3_)_2_NH_2_][Eu(TCPB)(H_2_O)_2_]·DMF}_n_	PO_4_^3−^, HPO_4_^2−^		0.139 mM			water	[90]
Zeolitic Imidazolate Frameworks (ZIFs)								
24	ZIF-90a	CrO_4_^2−^	5.03 × 10^3^		829.2341		water	[91]
25	M-ZIF-90	CN^−^	3.3 × 10^5^	2 μM			water	[92]
26	nano-ZIF-8	ClO^−^ and SCN^−^		0.133 nM 0.204 nM		5	water	[93]
27	MAPbBr_3_@ZIF-8	ClO^−^	0.141 × 10^6^	31.9 nM	207.9		water	[94]
Covalent-Organic Frameworks (COFs)								
28	BCMP-3	F^−^			950	5	THF	[95]
29	TFPPy-DETHz-COF	F^−^		50.5 ppb	1090		water	[96]
30	Fe-CTF	F^−^		5 nM			water	[97]
31	COF-TT	CrO_4_^2−^, Cr_2_O_7_^2−^, and MnO_4_^−^	1.4 × 10^4^, 1.4 × 10^4^, and 1.5 × 10^4^	3.43 × 10^−4^ M, 3.43 × 10^−4^ M, and 3.20 × 10^−4^ M	446		water	[98]
32	PP-TzDa	MnO_4_^−^	3.279 × 10^3^	0.01 mM	583		water	[99]
33	DATG_Cl_-iCONs	F^−^	2.25 × 10^3^	5 ppb	155	5	water	[100]
34	UiO@COF1	PO_4_^3−^		0.067 μM	504		water	[101]
35	IC-COF	PO_4_^3−^ and CO_3_^2−^	3.5 × 10^3^ and 3.1 × 10^3^	0.61 μM 1.2 μM	647	5	water	[102]
36	TFHPB-TAPB-COF	F^−^	3.2 × 10^4^	1592 × 10^−9^ M	751		water	[103]
37	TFHPB-TTA-COF	F^−^	3.4 × 10^4^	1125 × 10^−9^ M	1472		water	[103]
38	ACOF	F^−^	1.2 × 10^4^	2.5 μM	674	3	water	[104]

## Data Availability

No data are contained within the article. Additional information is available on request from the corresponding author.

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
