# Peer review of "Reticular Chemistry for Optical Sensing of Anions"

_ijms, 2023, doi:10.3390/ijms241713045_

Round 1
Reviewer 1 Report
The text highlights the importance of reticular chemistry in the last decades as a field of porous crystalline molecular materials. It describes researchers' efforts to create an ideal platform for the analysis of various anions based on optical detection techniques (chromogenic and fluorogenic). A positive aspect of the text is that it emphasizes the advantages of materials based on reticular chemistry, such as Metal-Organic Frameworks (MOFs), Zeolitic Imidazolate Frameworks (ZIFs), and Covalent-Organic Frameworks (COFs).
Could the authors add more information in the introduction about the challenges and limitations associated with using reticular chemistry for anion detection? What are they and how can they be overcome? How do the sensitivity and selectivity of these crosslinking materials compare with other currently available anion detection techniques?
The authors could make a comparison of sensitivity and selectivity of these reticular materials with other existing detection techniques for anions, the text should include more comprehensive and up-to-date references. The authors should consider enriching the bibliography with recent papers in the field to support their claims and provide a thorough literature review.
A review is done by authors with experience in the fields they are reviewing. Thus, the authors should add and discuss their own references/works related to the subject of the review type work done which outline the motivation of the research/study.
The bibliography in this manuscript is rather poor (62 Refs. for a review) and should be enriched at least with the recent papers in the field. Therefore, we do not agree with the statements of the authors of Abstract (“In the last few decades, reticular chemistry has grown significantly as a field of porous crystalline molecular materials” and from the line 99 “This review tried to summarize “.
Authors should add an introduction (generalities) for each subsection (e.g. line 338 etc.) and not go directly to reporting what other authors have done.
Also, at the end of each subsection, important findings need to be provided at the end of this section in terms of the most efficient/important MOF, ZIF and COF systems.
Be careful when processing figure 10a, you can't see the whole word.
Minor points: line 49: (10)=[10].; line 197: drinking = drinking water; line 287: act=acted; line 440: phosphate ion = phosphate ions.
The text highlights the importance of reticular chemistry in the last decades as a field of porous crystalline molecular materials. It describes researchers' efforts to create an ideal platform for the analysis of various anions based on optical detection techniques (chromogenic and fluorogenic). A positive aspect of the text is that it emphasizes the advantages of materials based on reticular chemistry, such as Metal-Organic Frameworks (MOFs), Zeolitic Imidazolate Frameworks (ZIFs), and Covalent-Organic Frameworks (COFs).
Could the authors add more information in the introduction about the challenges and limitations associated with using reticular chemistry for anion detection? What are they and how can they be overcome? How do the sensitivity and selectivity of these crosslinking materials compare with other currently available anion detection techniques?
The authors could make a comparison of sensitivity and selectivity of these reticular materials with other existing detection techniques for anions, the text should include more comprehensive and up-to-date references. The authors should consider enriching the bibliography with recent papers in the field to support their claims and provide a thorough literature review.
A review is done by authors with experience in the fields they are reviewing. Thus, the authors should add and discuss their own references/works related to the subject of the review type work done which outline the motivation of the research/study.
The bibliography in this manuscript is rather poor (62 Refs. for a review) and should be enriched at least with the recent papers in the field. Therefore, we do not agree with the statements of the authors of Abstract (“In the last few decades, reticular chemistry has grown significantly as a field of porous crystalline molecular materials” and from the line 99 “This review tried to summarize “.
Authors should add an introduction (generalities) for each subsection (e.g. line 338 etc.) and not go directly to reporting what other authors have done.
Also, at the end of each subsection, important findings need to be provided at the end of this section in terms of the most efficient/important MOF, ZIF and COF systems.
Be careful when processing figure 10a, you can't see the whole word.
Minor points: line 49: (10)=[10].; line 197: drinking = drinking water; line 287: act=acted; line 440: phosphate ion = phosphate ions.
Author Response
ijms-2545077
Reviewer 1
We appreciate the reviewer’s overall positive appraisal of our work. The questions/comments posed certainly led to the improvement of our manuscript.
- Could the authors add more information in the introduction about the challenges and limitations associated with using reticular chemistry for anion detection? What are they and how can they be overcome? How do the sensitivity and selectivity of these crosslinking materials compare with other currently available anion detection techniques?
Answer: The challenges and limitations have already been added in the last paragraph of the introduction. The advantages of reticular chemistry in the optical sensing of anions as compared to other methods of sensing have already been add in the third paragraph of the introduction.
- The authors could make a comparison of sensitivity and selectivity of these reticular materials with other existing detection techniques for anions, the text should include more comprehensive and up-to-date references. The authors should consider enriching the bibliography with recent papers in the field to support their claims and provide a thorough literature review.
Answer: The comparison of the sensitivity and selectivity has been shown in Table 1. New and recent references have been added as 18e, 18f, 18g, 34, 38, 39, and 66
- A review is done by authors with experience in the fields they are reviewing. Thus, the authors should add and discuss their own references/works related to the subject of the review type work done which outlines the motivation of the research/study.
Answer: The paper published by the author on anion sensing with MOF has been added and discussed in the text as reference 34.
- The bibliography in this manuscript is rather poor (62 Refs. for a review) and should be enriched at least with the recent papers in the field. Therefore, we do not agree with the statements of the authors of Abstract (“In the last few decades, reticular chemistry has grown significantly as a field of porous crystalline molecular materials” and from the line 99 “This review tried to summarize “.
Answer: New and recent references have been added as 18e, 18f, 18g, 34, 38, 39, and 66. There are several references within one reference (such as 11, 12, 13, 14, 15, 16, 17, 18, 20, and 21) so the total reference count is much larger.
- Authors should add an introduction (generalities) for each subsection (e.g. line 338 etc.) and not go directly to reporting what other authors have done.
Answer: An introduction for each subsection has been added.
- Also, at the end of each subsection, important findings need to be provided at the end of this section in terms of the most efficient/important MOF, ZIF, and COF systems.
Answer: An conclusion for each subsection of MOF ZIFs and COFs has been added.
- Be careful when processing Figure 10a, you can't see the whole word.
Answer: Figure 10a has been corrected.
- Minor points: line 49: (10)=[10].; line 197: drinking = drinking water; line 287: act=acted; line 440: phosphate ion = phosphate ions.
Answer: All these points have been corrected.
Reviewer 2 Report
The review titled "Reticular Chemistry for Optical Sensing of Anions" explores the captivating subject of anion sensing using diverse porous materials such as MOFs and COFs. This topic holds tremendous potential and is important, with numerous advantages for both academic and industrial applications. The data presented in this manuscript is articulated with great precision, encompassing a wide range of porous materials. The review's content is thorough and comprehensive, especially regarding stable MOF materials and their advantages, along with impressive theoretical investigations.
After a careful evaluation of this manuscript, I recommend it for publication, provided that the following points are addressed:
- Since the review specifically focuses on anion sensing, the authors should include comparative tables of all reported MOFs and COFs for anion sensing separate tables, detailing their performance, stability, and selectivity for different anions. This addition would enhance the review's accessibility to a broader audience.
- While the review adeptly discusses the advantages of MOFs and porous materials for anion sensing, it is equally crucial to address their limitations in a separate paragraph. Particularly, the review should elaborate on the limitations of pristine porous materials/MOFs in terms of water stability and how this aspect has hindered progress in the field. Furthermore, the authors should provide a comprehensive discussion on potential future directions to overcome these limitations for anion sensing applications.
- In addition to the advantages of porous materials, it is vital to acknowledge the challenges and limitations they pose for anion sensing. The authors should dedicate a detailed paragraph to explain how MOFs may undergo structural decomposition under certain experimental conditions, especially concerning reductions. This clarification would be valuable in understanding the suitability of MOFs for sensing applications.
- To increase the visibility of this review, the authors are encouraged to cite the following relevant papers: ACS Appl. Nano Mater. 2019, 2, 8, 5169–5178, ACS Omega 2022, 7, 18, 15275–15295 and JACS, 2023 145 (17), 9850-9856. These papers either involve anions in the pores or have been tested for advanced applications.
- Although the review primarily covers well-known and state-of-the-art MOFs and COFs, it would greatly benefit from a detailed discussion of the theoretical understanding and structural aspects of these MOF materials. I recommend incorporating this data to provide a more comprehensive overview.
- It is widely recognized that MOFs offer distinct advantages over COFs concerning sensing capabilities, primarily attributed to their inherent interconnectivity and structural benefits. Therefore, the authors should provide a comprehensive commentary on this matter.
Author Response
Reviewer 2
We appreciate the reviewer’s overall positive appraisal of our work. The questions/comments posed certainly led to the improvement of our manuscript.
- Since the review specifically focuses on anion sensing, the authors should include comparative tables of all reported MOFs and COFs for anion sensing separate tables, detailing their performance, stability, and selectivity for different anions. This addition would enhance the review's accessibility to a broader audience.
Answer: The comparison of the sensitivity and selectivity has been shown in Table 1.
- While the review adeptly discusses the advantages of MOFs and porous materials for anion sensing, it is equally crucial to address their limitations in a separate paragraph. Particularly, the review should elaborate on the limitations of pristine porous materials/MOFs in terms of water stability and how this aspect has hindered progress in the field. Furthermore, the authors should provide a comprehensive discussion on potential future directions to overcome these limitations for anion sensing applications.
Answer: This point has already been discussed in the last paragraph of the introduction.
- In addition to the advantages of porous materials, it is vital to acknowledge the challenges and limitations they pose for anion sensing. The authors should dedicate a detailed paragraph to explain how MOFs may undergo structural decomposition under certain experimental conditions, especially concerning reductions. This clarification would be valuable in understanding the suitability of MOFs for sensing applications.
Answer: This point has already been discussed in the introduction section.
- To increase the visibility of this review, the authors are encouraged to cite the following relevant papers: ACS Appl. Nano Mater. 2019, 2, 8, 5169–5178, ACS Omega 2022, 7, 18, 15275–15295 and JACS, 2023 145 (17), 9850-9856. These papers either involve anions in the pores or have been tested for advanced applications.
Answer: These references have been added
- Although the review primarily covers well-known and state-of-the-art MOFs and COFs, it would greatly benefit from a detailed discussion of the theoretical understanding and structural aspects of these MOF materials. I recommend incorporating this data to provide a more comprehensive overview.
Answer: This review mainly focuses on the anion sensing and reticular chemistry. Discussion of theoretical aspect is beyond the scope of this review.
- It is widely recognized that MOFs offer distinct advantages over COFs concerning sensing capabilities, primarily attributed to their inherent interconnectivity and structural benefits. Therefore, the authors should provide a comprehensive commentary on this matter.
Answer: This point has already been discussed in the introduction and in the starting point of each subsection.
Round 2
Reviewer 1 Report
The work, in my opinion have considered my comments. This paper could be accepted for publication. The Authors have done a truly significant work on the amendments and additions to their original work.
Reviewer 2 Report
After incorporating the suggested data, the manuscript has noticeably improved in quality. I am happy to accept the review for publication.